# Korean Thistle (*Cirsium japonicum* var. *maackii* (Maxim.) Matsum.): A Potential Dietary Supplement against Diabetes and Alzheimer’s Disease

**DOI:** 10.3390/molecules24030649

**Published:** 2019-02-12

**Authors:** Aditi Wagle, Su Hui Seong, Srijan Shrestha, Hyun Ah Jung, Jae Sue Choi

**Affiliations:** 1Department of Food and Life Science, Pukyong National University, Busan 48513, Korea; aditiwagle05@gmail.com (A.W.); seongsuhui@naver.com (S.H.S.); srijanstha003@gmail.com (S.S.); 2Department of Food Science and Human Nutrition, Chonbuk National University, Jeonju 54896, Korea

**Keywords:** α-Glucosidase, BACE1, *Cirsium maackii*, Korean thistle, luteolin, supplements

## Abstract

In the search for natural products having a dual inhibitory action on diabetes and Alzheimer’s disease, this study investigated the activity of different parts of Korean thistle (*Cirsium japonicum* var. *maackii* (Maxim.) Matsum), and its fractional constituents by in vitro enzymatic and in silico molecular docking studies. *Cirsium maackii* has been used as a traditional medicine for the treatment of several diseases. The ethyl acetate and dichloromethane fractions of a leaf extract showed α-glucosidase and BACE1 inhibitory activity, respectively. Furthermore, the isolated compound, luteolin, exhibited concentration-dependent non-competitive inhibition against both α-glucosidase and BACE1 (IC_50_ = 51.27 ± 1.23 and 13.75 ± 0.26 μM; *K_i_* value = 52.04 and 14.76 μM, respectively). Moreover, docking studies showed that luteolin formed a strong hydrogen bond with the peripheral binding amino acid residues, and hydrophobic interactions with the α-glucosidase and BACE1 enzymes. Therefore, Korean thistle may act as an important dietary supplement against diabetes and Alzheimer’s disease, especially the leaves, because of the preponderance of the active component, luteolin, making Korean thistle a promising candidate for more detailed in vitro and in vivo studies.

## 1. Introduction

Diabetes is one of the most common metabolic disorder in today’s world. According to the International Diabetes Federation, 451 million people were affected with diabetes in 2017 and that number might increase to 693 million by the year 2045. Moreover, it is estimated that nearly half of all the population are undiagnosed with diabetes, which could increase their risk of developing more complications [1]. The pathogenesis of type 2 diabetes (T2D) has several causes, which include heredity, gene mutations, and obesity [2,3]. One way to decrease post-prandial hyperglycemia in T2D is to decrease the absorption of carbohydrates from the food we consume. This can be done by inhibiting the carbohydrate-hydrolyzing enzyme α-glucosidase present in the epithelium of the small intestine. α-Glucosidase, an exo-type carbohydrase enzyme widely distributed in microorganisms, plants, and animal tissues, catalyzes the liberation of α-glucose from the non-reducing end of the substrate [4]. Inhibiting α-glucosidase decreases the rate of hydrolytic cleavage of oligosaccharide, and the process of carbohydrate digestion extends to the lower part of the small intestine, thus delaying the overall absorption rate of glucose into the blood [5,6].

Likewise, Alzheimer’s disease (AD) is a degenerative brain disease characterized by a gradual decline in the cognition and behavior functions affecting day-to-day activities. According to the Alzheimer’s Association (2018), in the U.S. approximately 5.7 million people have AD. The prevalence is increasing so rapidly that among every 10 people aged 65 or older, one is likely to have AD, and by the year 2050, the figure may increase to 88 million [7]. Though different studies have been performed regarding the treatment of AD, the search for a treatment still continues.

Although the exact mechanism for AD remains unclear, several hypotheses have been proposed including the amyloid β-protein (Aβ) cascade, the tau hypothesis, the inflammation hypothesis, and the cholinergic and oxidative stress hypothesis [8]. According to the amyloid hypothesis, the triggering of different pathologic cascades by amyloid β-protein (Aβ) may lead to neurodegeneration. Aβ is formed from the cleavage of the amyloid precursor protein (APP) by proteases: namely, the β-site amyloid precursor protein cleaving enzyme 1 (β-secretase, BACE1) and γ-secretase [9]. Proteases are among the prime therapeutic targets in AD. Accumulation of Aβ within neural tissue may cause an imbalance between Aβ production and clearance, leading to the aggregation of Aβ and plaque formation, and eventually leading to the formation of neurofibrillary tangles [10]. Thus, BACE1 inhibition is one of the potential targets for decreasing Aβ levels in AD.

Since T2D and AD are common in the elderly, they may be co-related. Though the exact mechanisms are still unclear, causes of T2D such as insulin resistance [11], obesity [12], and several other components of the metabolic syndrome may also cause AD. Also, deficiency of or resistance to insulin, impaired growth factor signaling, advanced glycation end products and their receptors, glucose toxicity, vascular injury, and cerebrovascular events are among the proposed mechanisms for a link between T2D and AD [13]. Finding treatments for T2D and AD is an ongoing challenge, regardless of the efforts made. Therefore, knowing the key enzymes associated with T2D and AD, α-glucosidase and BACE1 respectively, warrants a search for potentially safer natural inhibitors, giving motivation to explore biologically active compounds from a highly diverse group of plants.

*Cirsium japonicum* var. *maackii* (Maxim.) Matsum. (synonym = *C. maackii*), or Korean thistle, is a perennial herbaceous medicinal plant belonging to the family Compositae. Thistles are found throughout the world, however, the plant is widely distributed in the mountains and fields of Korea, China, and Japan. It has been widely used as a traditional herbal medicine and is also listed in the Korean, Chinese and Japanese pharmacopoeias for treating hemorrhages, hypertension, hepatitis and ureteral disorders [14,15]. Several medicinal benefits of *C. maackii* have been reported, which include anti-inflammatory [16], hepatoprotective [17], and anti-cancer [18] effects. Additionally, it can be used to prevent diabetic complications and oxidative stress-related diseases through its inhibitory activity against advanced glycation end-product formation [19] and aldose reductase inhibitory activity [20,21]. However, its anti-diabetic and anti-AD activities have not been studied in detail. Furthermore, dietary supplements can contribute to improved health. In particular, plants with biological activities can have added benefits and can be consumed as food ingredients which can help in preventing those diseases. Accordingly, we focused our study on the active components of *C. maackii* (Figure 1) as dietary supplements for treating T2D and AD through the inhibition of the α-glucosidase and BACE1 enzymes, respectively.

## 2. Results

### 2.1. α-Glucosidase and BACE1 Inhibitory Activity of Parts/Extracts and Solvent Soluble Fractions of C. maackii

The aerial part of *C. maackii* showed high inhibitory activity against α-glucosidase and BACE1 with half maximal inhibitory concentration (IC_50_) values of 375.66 ± 3.21 and 41.43 ± 0.23 µg/mL, respectively (Table 1). Likewise, α-glucosidase and BACE1 assays for the methanol (MeOH) extracts of root, stem, flower, and leaves were carried out. Among them, the leaf extract showed a concentration-dependent inhibition of both α-glucosidase and BACE1, with IC_50_ values of 55.07 ± 3.64 and 73.98 ± 0.06 µg/mL, respectively (Figure 2A,B). Similarly, in order to find the active fraction, four fractions, namely dichloromethane (CH_2_Cl_2_), ethyl acetate (EtOAc), *n*-butanol (*n*-BuOH), and water (H_2_O) extracts, were assayed. The EtOAc fraction showed potent α-glucosidase inhibitory activity, with an IC_50_ value of 32.79 ± 1.63 µg/mL, while CH_2_Cl_2_ followed by EtOAc fractions showed activity against BACE1 with IC_50_ values of 25.77 ± 0.57 and 39.88 ± 1.66 µg/mL, respectively. Acarbose and quercetin, the positive controls, showed an inhibition of 64.43% at a concentration of 250 µg/mL and 70.01% at a concentration of 4.5 µg/mL for α-glucosidase and BACE1, respectively.

### 2.2. Pro-Oxidant Activity of Different Compounds from C. maackii

The isolated compounds were tested against pro-oxidant activity which were depicted in Figure 3. Our result demonstrated that luteolin have lower reducing power than positive control ascorbic acid. While in case of its glucosides, luteolin 5-*O*-*β*-d-glucopyranoside exhibited higher reducing power.

### 2.3. Anti-Oxidant Activity of Different Compounds from C. maackii

Among the isolated compounds, the anti-oxidant activity [2,2-diphenyl-1-picrylhydrazyl (DPPH) radical scavenging and peroxynitrite (ONOO^−^) scavenging activity] were carried out as shown in Table 2. All these compounds showed significant inhibition of DPPH radical when compared to that of positive control, ascorbic acid (21.35 ± 0.09 µM). Additionally, luteolin (3.05 ± 0.06 µM), an aglycoside moiety exhibited powerful DPPH scavenging ability than its glycosides. Moreover, for ONOO^−^ scavenging activity, both luteolin and luteolin 5-*O*-*β*-d-glucopyranoside demonstrated significant scavenging activity with half maximal effective concentration (EC_50_) values of 0.86 ± 0.04 and 0.50 ± 0.03 µM, respectively. Penicillamine, the positive control used for the experiment showed an EC_50_ of 2.28 ± 0.02 µM.

### 2.4. Anti-Diabetic and Anti-AD Activity of Different Compounds from C. maackii

Among the isolated compounds luteolin, luteolin 5-*O*-*β*-d-glucopyranoside, and luteolin 7-*O*-*β*-d-glucopyranoside anti-diabetic (α-glucosidase and PTP1B) and anti-AD (BACE1) activity assays were carried out (Table 3). The aglycone moiety, i.e., luteolin, exhibited potent α-glucosidase and BACE1 inhibitory activity with IC_50_ values of 51.27 ± 1.23 and 13.75 ± 0.26 µM, respectively. In the case of α-glucosidase, the glucosides luteolin 5-*O*-*β*-d-glucopyranoside and luteolin 7-*O*-*β*-d-glucopyranoside demonstrated moderate inhibition, having IC_50_ values of 270.03 ± 4.69 and 248.77 ± 2.56 µM, respectively. The results of the glucosides were comparable to the positive control, acarbose (IC_50_ = 267.27 ± 2.29 µM). In contrast, luteolin glucosides had weak inhibition against BACE1. No inhibition of PTP1B was observed with the compounds at concentrations up to 400 µM.

### 2.5. Enzyme Kinetic Analysis of Compounds with α-Glucosidase and BACE1

To analyze the types of α-glucosidase and BACE1 inhibition exerted by the isolated compounds, enzyme kinetic analysis was performed using Lineweaver-Burk and Dixon plots (Table 3 and Figure 4 and Figure 5). Luteolin inhibited both the α-glucosidase and BACE1 enzymes in a non-competitive manner, as all the lines crossed the negative portion of the *X*-axis and the value of *V_max_* decreased, whereas −1/*K_m_* (horizontal axis intercept) was constant with increasing concentrations of inhibitors (Figure 4A and Figure 5A). Likewise, the increasing concentrations of substrate resulted in a family of lines that shared a common intercept on the left of the vertical axis and above the horizontal axis (Figure 4C), indicating a decrease in *V_max_* and an increase in *K_m_*. These results in turn indicated that luteolin 5-*O*-*β*-d-glucopyranoside caused a mixed type of inhibition towards α-glucosidase enzymes, involving a combination of partially competitive and pure non-competitive inhibition. Moreover, in the case of luteolin 7-*O*-*β*-d-glucopyranoside (Figure 4E), the sets of Lineweaver-Burk plot were, in all instances, typical of competitive inhibition because the linear graphs of each set coincide on the *Y*-axis. According to the Dixon plot, the *K_i_* values of the compound were 52.04, 271.80, and 251.20 µM for luteolin, luteolin 5-*O*-*β*-d-glucopyranoside, and luteolin 7-*O*-*β*-d-glucopyranoside, respectively (Figure 4B,D,F) in α-glucosidase and 14.76 µM for luteolin in BACE1 (Figure 5B).

### 2.6. Molecular Docking Simulation of α-Glucosidase and BACE1 Inhibition

To clarify how the bioactive compounds interacted with the active sites of the enzymes, molecular docking simulation was performed for the isolated compounds, namely luteolin, luteolin 5-O-β-d-glucopyranoside, and luteolin 7-O-β-d-glucopyranoside. The binding energies of luteolin and its glucosides with the interacting residues, together with H-bond interactions, hydrophobic interacting residues, and numbers of H-bonds are listed in Table 4 and Table 5.

For α-glucosidase, the modelling results suggest that luteolin formed five H-bonds with Glu271, Ile272, Ser298, and Thr274, with a binding energy of −7.25 kcal/mol. In addition, the α-glucosidase-luteolin complex was stabilized by hydrophobic interactions with Gly269, Arg270, Glu296, Asn259, Ile262, Val266, and Arg263 (Figure 6A,B). Furthermore, luteolin 5-*O*-*β*-d-glucopyranoside had binding energies of −7.22 and −6.98 kcal/mol as a catalytic and an allosteric inhibitor, respectively, exhibiting mixed inhibition. It formed seven H-bonds with the catalytic active site (His112, Asp215, Gln182, Tyr158, Glu411, Asp307, and Gln353) and six H-bonds with the allosteric site (Thr290, Ser298, Ile272, Arg270, and His295) (Figure 6C–F). Likewise, the α-glucosidase-luteolin 7-*O*-*β*-d-glucopyranoside complex had a binding energy of −7.21 kcal/mol, forming six H-bonds with Asp69, Arg442, Pro312, Asp242, and His280. In addition, hydrophobic interactions were also observed between luteolin 7-*O*-*β*-d-glucopyranoside and the Phe159, Glu277, Asp352, Phe314, Phe303, Arg315, Leu313, Tyr158, Gln279, Val216, Asp215, and Phe178 residues (Figure 6G,H). As shown in Table 4, the docking results showed that acarbose (a catalytic inhibitor) formed 17 hydrogen bonds with Gln182, Asp69, Asp215, Arg213, Glu277, Asp352, Arg442, Asp307, His280, Asp242, Ser240, and Tyr158, whereas BIP (an allosteric inhibitor) formed two hydrogen bonds with Glu296 and His295. Similarly, in the case of BACE1, luteolin formed three H-bonds with Ser299 and Gln304 along with the hydrophobic interacting residues Ser10, Gly11, Gln12, Gly13, Pro308, Val170, Thr232, Gly230, Ala335, Tyr14, Val336, Glu339, and Arg307 (Figure 7A,B), with a binding energy of −7.18 kcal/mol. Luteolin 5-*O*-*β*-d-glucopyranoside and luteolin 7-*O*-*β*-d-glucopyranoside demonstrated lower binding energies: −6.02 and −5.47 kcal/mol, respectively (Figure 7C–F). At the same time, luteolin 5-*O*-*β*-d-glucopyranoside formed six H-bonds with Asp32, Asp228, Ile126, and Phe108, and luteolin 7-*O*-*β*-d-glucopyranoside formed nine H-bonds with Lys107, Phe108, Lys224, Tyr198, Thr329, Arg235, Asp228, and Thr231. The known catalytic inhibitor QUD formed four H-bonds with Asp32, Asp228, and Gly230, with a binding energy of −9.30 kcal/mol, whereas PMF, an allosteric inhibitor, formed one H-bond with Ser10, with a binding energy of −6.15 kcal/mol.

### 2.7. Absorption, Distribution, Metabolism, and Excretion (ADME) Properties of Different Compounds from Korean Thistle, C. maackii

ADME properties such as human intestinal absorption (HIA), plasma protein binding, and cell permeability using Madin-Darby canine kidney (MDCK) and Caco-2 cells were evaluated using tool PreADMET (Table 6). Our results showed luteolin to have strong human intestinal absorption while its glucosides showed moderate absorption. Likewise, luteolin strongly bound to plasma protein while its glucoside showed weak binding. Furthermore, luteolin exhibited moderate permeability for both MDCK and Caco-2 cells.

## 3. Discussion

T2D and AD affect many people around the world. Their prevalence is increasing so rapidly that a search for novel treatments is a major focus of research. T2D and AD are considered age-related diseases, associated with a genetic predisposition and with similar pathological features in islet cells and the brain, respectively. Though several studies have revealed that patients with T2D are at high risk of developing AD, the exact mechanism is ambiguous. The common features of T2D and AD are insulin resistance, abnormal protein processing, oxidative stress, high cholesterol level, formation of advanced glycation end products, association with cardiovascular disease, blood vessel abnormalities, and the activation of inflammatory pathways [2,3,22]. AD is manifested by the progressive loss of memory and decline in cognitive function, which are associated with the formation of Aβ protein, neurofibrillary tangles consisting of intracellular abnormally phosphorylated tau protein, with loss of neurons and synapses in the hippocampus and cortex regions [23,24]. In T2D, the underlying mechanism includes insufficient insulin secretion from pancreatic β-cells and insulin resistance. Initially, in response to the insulin resistance, pancreatic β-cells increase insulin secretion, leading to hyperinsulinemia. However, when β-cell function decreases, insulin production is inadequate to overcome insulin resistance, causing the blood glucose level to rise, leading to T2D [13]. In addition, to regulate glucose homeostasis, insulin has also been found to play a major role in regulating APP and its derivative Aβ protein associated with senile plaques, a neuropathological hallmark of AD [25]. Though several remedies are available worldwide for the treatment of these diseases, the search is ongoing for new natural components as an alternative with fewer adverse effects. And furthermore, dietary supplements can play a major role in the prevention of these diseases.

In our present study, a preliminary screening assay demonstrated that the aerial part and MeOH extract of *C. maackii* showed moderate inhibitory effects against α-glucosidase and BACE1, which prompted us to further investigate its different parts which were responsible for the activity. Among the different parts tested (root, stem, leaves, and flower), the leaves showed a concentration-dependent inhibition against both α-glucosidase and BACE1 (Figure 2A,B). However, the other parts did not exhibit prominent activity up to concentration of 1000 µg/mL for α-glucosidase. Though the major active compounds isolated from *C. maackii*, i.e., luteolin and luteolin 5-*O*-*β*-d-glucopyranoside were found in the root and stem, including a small amount in the flower as well [20], only the leaves exhibited significant activity. The reason might be the presence of the other compounds found in the root and stem that antagonize the activity of the active compounds. At the same time, for BACE1, the leaves preceded by the flower part had the next highest activity level, with an inhibition of 52.11% at a concentration of 100 µg/mL, followed by the stem and root, which showed an intermediate level of inhibition against BACE1. Furthermore, the high activity of the leaves due to their abundance of active constituents encourages us to find out which fractions to be responsible for their activity. Therefore, the partitioning of the MeOH extract leads to the four solvent soluble fractions where the separation of the components was based on their polarity. And among the four fractions, the EtOAc fraction showed the significant inhibitory activity towards α-glucosidase; while the CH_2_Cl_2_, followed by the EtOAc and the *n*-BuOH fractions, showed the most potent inhibition of BACE1. The CH_2_Cl_2_ fraction was found to have abundant amount of lupeol and lupeol acetate, and a previous study had shown that lupeol had potent BACE1 inhibitory activity [26]. These results can explain the greater activity of the CH_2_Cl_2_ fraction compared with that of EtOAc fraction and the other non-polar components. Thus, this outcome minimizes the inconvenience of using the whole plant and confirms the usefulness of specific parts, i.e., the leaves of *C. maackii*, as potential dietary supplements for anti-T2D and anti-AD activity.

Furthermore, to find whether the isolated compounds have pro-oxidant and/or anti-oxidant effect or not, different procedures were followed (Table 2 and Figure 3). As we can see in Figure 3, there were increase in the reducing power of the compounds with increase in concentrations. However, luteolin which possess the strong anti-oxidant activity possess low reducing power implying that its reducing power might not aid to its anti-oxidative activity. Moreover, the greater will be its reducing power, the more is its absorbance value. Nevertheless, the anti-oxidant activity of the compounds were similar to that of the study previously reported by Jung et al. [20].

Likewise, among the isolated compounds luteolin, luteolin 5-*O*-*β*-d-glucopyranoside, and luteolin 7-*O*-*β*-d-glucopyranoside, luteolin exhibited five-fold more potent inhibitory activity against α-glucosidase compared with the positive control, acarbose, while its glucosides, namely luteolin 5-*O*-*β*-d-glucopyranoside and luteolin 7-*O*-*β*-d-glucopyranoside, had inhibitory effects similar to acarbose. At the same time, against BACE1, luteolin exhibited stronger inhibitory activity compared to that of its glucosides. These results highlight that the aglycone part of the flavonoid exhibited more inhibitory activity than its glucosides. Nevertheless, no notable differences were observed between the 5th and 7th glucose-substituted luteolin. Moreover, as we defined the effectiveness levels of the tested compounds, we aimed to evaluate the type of α-glucosidase and BACE1 inhibition by luteolin and its derivatives. Lineweaver-Burk and Dixon plots were employed in order to investigate the inhibition kinetics. As observed in Figure 4 and Figure 5, luteolin was a non-competitive inhibitor of α-glucosidase and BACE1, signifying the equal affinity to the free enzyme as well as the enzyme–substrate complex. A non-competitive inhibitor binds to an enzyme whether the substrate is at a low or high concentration [27]. Likewise, luteolin 5-*O*-*β*-d-glucopyranoside is a mixed type inhibitor for α-glucosidase (Figure 4), indicating the substrate or the inhibitor binding affect its enzyme affinity. In mixed type inhibition, as inhibitor and substrate can together be attached to the enzyme as inhibitor does not bound in the active site of the enzyme. However, the substrate binding affinity can be declined when the inhibitor is present. Additionally, luteolin 7-*O*-*β*-d-glucopyranoside showed competitive inhibition towards α-glucosidase, which implies that luteolin 7-*O*-*β*-d-glucopyranoside competes with the substrate for the active site (Figure 4). The *K_i_* values can also be a useful tool to compare the activity of the compounds, showing the binding affinity of the inhibitor to the enzyme (Table 3). These results suggest that luteolin exhibits the strongest binding affinity for the α-glucosidase and BACE1 enzymes.

Correspondingly, Ha et al. [28] also found non-competitive α-glucosidase inhibition by luteolin, with an IC_50_ of 45.4 μM, which is similar to our findings. Furthermore, Yan et al. [29] demonstrated that luteolin reversibly inhibited α-glucosidase in a non-competitive manner, with an IC_50_ value of (1.72 ± 0.05) × 10^−4^ mol L^−1^. Moreover, Li et al. [30] also observed a potent in vitro activity against the enzyme, α-glucosidase. Also, Tadera et al. [31] observed that the inhibition percentage of luteolin at 200 µM was 92%, with mixed type (close to non-competitive) inhibition. On the other hand, Kim et al. [32] showed that luteolin inhibited α-glucosidase by 36% at a concentration of 0.5 mg/mL and was stronger than acarbose and luteolin 7-*O*-*β*-d-glucopyranoside. In addition, luteolin isolated from *Scutellaria lateriflora* weakly inhibited sucrase [33]. Moreover, Proença et al.’s study on the structure activity relationship of flavonoids against α-glucosidase showed that luteolin was a significant inhibitor, with an IC_50_ value of 46 ± 6 µM [34]. In addition, Xu et al. [35] found that luteolin showed mixed type inhibition against α-glucosidase, with an IC_50_ value of 18.6 μM. Also, Choi et al. demonstrated that luteolin exerted non-competitive inhibition against BACE1, with an IC_50_ of 5.0 × 10^−7^ M and a *K_i_* value of 6.2 × 10^−5^ M [36]. Likewise, luteolin had anti-amnesic and neurovascular protective effects in the study performed by Liu et al. [37]. Rezai-Zadeh et al. found that luteolin inhibited Aβ_1–40,42_ generations from SweAPP N2a cells and Tg2576 mouse-derived primary neuronal cells [38]. Furthermore, luteolin also reduced GSK-3α/β activation and altered PS1 processing/phosphorylation in SweAPP N2a cells. Moreover, luteolin improved the cognitive function in Aβ_1–40_ induced amnesic rats via antioxidant activation [39]. Cheng et al. also reported the neuroprotective activity of luteolin through inhibition of the protein levels of JNK, ERK and p38 MAP kinases and caspase 3 activations [40]. The discrepancies found in different studies may be due to the use of different enzyme and substrate concentrations, enzymes from different origins (yeast and rat intestinal source for α-glucosidase), and incubation times.

In the nutraceutical field, molecular docking techniques have been widely used to predict ligand–target binding affinity and to better understand the molecular basis of the biological responses. Therefore, molecular docking simulations were performed with the bioactive compounds luteolin, luteolin 5-*O*-*β*-d-glucopyranoside, and luteolin 7-*O*-*β*-d-glucopyranoside isolated from the leaves of *C. maackii*. The results are shown in Table 4 and Table 5. For α-glucosidase, the docking result suggests that the 3′-OH group of luteolin formed hydrogen bonds with Ile272, Ser298, Thr274, and the 4′-OH group with Thr274. Furthermore, the 7-OH group of luteolin formed a strong hydrogen bond with Glu271. Also, the hydrophobic interactions with Gly269, Arg270, Glu296, Asn259, Ile262, Val266, and Arg263 aided in stabilizing the complex. Additionally, the lower binding energy (−7.25 kcal/mol) of luteolin signifies the strong interactions with α-glucosidase. Besides, the mixed inhibition exhibited by luteolin 5-*O*-*β*-d-glucopyranoside was supported by the docking study, which showed its interaction with both the catalytic and allosteric residues. For the catalytic inhibition of luteolin 5-*O*-*β*-d-glucopyranoside, different hydroxyl groups of the sugar moiety formed strong H-bond interactions with Tyr158, Glu411, and Asp307, while the 7-OH formed an interaction with Gln353. Likewise, the 3′-OH interacted with Gln182 and 4′-OH with Asp215 and His112. And also, for allosteric inhibition, along with the glucose moiety, the 4′-OH group and the keto group at the 4th position formed hydrogen bonds with His295, along with the oxygen atom in the 1st position bonding with Ser298. As well, luteolin 7-*O*-*β*-d-glucopyranoside interacted with the active catalytic residues Asp69, Arg442, Pro312, Asp242, and His280. Moreover, the hydroxyl groups of the sugar moiety formed hydrogen bonds with Pro312 and Asp242. Likewise, the oxygen atom linked with the glucose moiety formed a hydrogen bond with His280. Also, 3′-OH formed an H-bond with Arg442 and Asp69, stabilizing the ligand–protein complex. In the case of BACE1, the 3′ and 4′-OH groups of luteolin at ring B formed H-bonds with Ser229, as did the 7-OH group with Gln304. Additionally, Ser10, Gly11, Gln12, Gly13, Pro308, Val170, Thr232, Gly230, Ala335, Tyr14, Val336, Glu339, and Arg307 interactions supported the luteolin-BACE1 complex. Consequently, the glucose OH moiety of luteolin 5-O-β-d-glucopyranoside formed two H-bonds with Phe108. In addition, the 7-OH group interacted with Asp228 and Asp32, and the 3′-OH and 4′-OH with Ile126. Subsequently, seven H-bonds were formed by the OH group of the glucose moiety of luteolin 7-O-β-d-glucopyranoside with Thr231, Asp228, Arg235, Thr329, Tyr198, and Lys224. Moreover, the 3′-OH and 4′-OH groups also formed H-bonds with Phe108 and Lys107. As shown in Table 4 and Table 5, the bulkier molecules, i.e., the luteolin glucosides, formed more H-bonds than did luteolin. The reason could be the availability of the hydroxyl group of the sugar moiety of the glucosides, which could readily interact with the residues of different amino acids of α-glucosidase and BACE1. Furthermore, though the glucoside molecules exhibited negative binding energies, energy might be required for the twisting of the molecules into the site, which would weaken their inhibition of the enzymes. The lower energy of luteolin compared to its glucosides would lead to higher binding affinity of the compounds towards the enzymes. Therefore, in silico studies provide additional insights into the underlying mechanism of action and binding mode of active compounds against metabolic key enzymes, supporting the experimental data. Nevertheless, additional in vivo studies are needed to strengthened the in vitro results.

In addition, the pharmacokinetic parameters of compounds isolated from *C. maackii* were estimated (Table 6). From the predicted data, we observed that the low molecular weight aglycone moiety i.e., luteolin exhibited the best pharmacokinetic properties, followed by luteolin 7-*O*-*β*-d-glucopyranoside and luteolin 5-*O*-*β*-d-glucopyranoside. As we know, predicting human intestinal absorption (HIA) of drugs is very important in the design, optimization and selection of oral drugs. HIA data are the sum of bioavailability and absorption evaluated from the ratio of excretion or cumulative excretion in urine, bile and feces [41]. This can be evaluated as follows: HIA between 0% and 20% indicates poorly absorbed compounds, HIA between 20% and 70% as moderately absorbed compounds and HIA between 70% and 100% as well absorbed compounds [42]. Luteolin, with its aglycone moiety, was well absorbed by human intestinal cells, while its glucosides were moderately absorbed. Moreover, the drugs that are prescribed for different treatments can either bind reversibly to the plasma proteins and lipids, called plasma protein binding, which is useful in predicting the therapeutic dose in clinical trials. The analysis of plasma protein binding was carried out based on the following criteria: chemicals strongly bound had a value more than 90% and weakly bound had a value less than 90%. Our result showed luteolin to be strongly bound to the plasma proteins. Furthermore, for the prediction of oral drug absorption, the MDCK cell and Caco-2 cell models have been recommended as a reliable in vitro model used in the drug selection process for measuring the intestinal absorption of drug candidates [43]. The analysis can be done using the criteria of low permeability being less than 4 nm/s, middle permeability 4–70 nm/s, and high permeability more than 70 nm/s [43,44]. In both cell models, luteolin exhibited moderate permeability compared to its glucosides, except for Caco-2 cells for which luteolin 7-*O*-*β*-d-glucopyranoside showed comparable permeability.

Several studies have highlighted the metabolic fate of luteolin and its glucosides. Shimoi et al. [45] revealed that luteolin 7-O-β-d-glucopyranoside was absorbed after hydrolysis to luteolin, which was changed to glucuronides during passage through the intestinal mucosa, and the monoglucuronide of the unchanged aglycone was the main conjugated metabolite circulating in the blood. Spencer et al. also found similar results [46]. Additionally, Kure et al. [47] confirmed luteolin 3′-*O*-glucuronide to be the major metabolite of luteolin after the oral administration of aglycone or its glucosides, with detectable amounts of luteolin 4′-*O*-glucuronide and luteolin 7-*O*-glucuronide in the liver, kidney, and small intestine. Also, Yasuda et al. [48] suggested that luteolin and luteolin monoglucoside were rapidly absorbed and circulated in the form of luteolin, monoglucoside, and monoglucuronide, which are responsible for exerting different biological functions. Interestingly, as flavonoid glucosides may serve as “pro-drugs” that subsequently release their respective aglycones in the gastrointestinal tract [49], luteolin 5-*O*-*β*-d-glucopyranoside and luteolin 7-*O*-*β*-d-glucopyranoside, along with luteolin with its aglycone moiety, might have biological activity towards α-glucosidase.

## 4. Materials and Methods

### 4.1. Chemicals and Reagents

Yeast α-glucosidase, *p*-nitrophenyl α-d-glucopyranoside (*p*-NPG), acarbose, *p*-nitrophenyl phosphate (*p*-NPP), ursolic acid, ethylenediaminetetraacetic acid (EDTA), ascorbic acid, penicillamine, and 2,2-diphenyl-1-picrylhydrazyl (DPPH) were purchased from Sigma Aldrich (St. Louis, MO, USA). PTP1B (human recombinant) was purchased from Biomol^®^ International, LLP (Plymouth Meeting, PA, USA), and dithiothreitol (DTT) was purchased from Bio-Rad Laboratories (Hercules, CA, USA). A BACE1 FRET assay kit (β-secretase) was purchased from PanVera Corp. (Madison, WI, USA). Peroxynitrite was purchased from Cayman Chemical (Ann Arbor, MI, USA). All other chemicals and solvents were purchased from E. Merck (Darmstadt, Germany), Honeywell Fluka (Morris Plains, NJ, USA), and Sigma-Aldrich, unless otherwise stated.

### 4.2. Plant Material

*C. maackii* plant was collected in Ulsan, Republic of Korea, in August 2001. The authentication of the plant specimen was done by a specialist in *Cirsium* taxonomy, Dr. Y. Kadota, at the Department of Botany, National Museum of Nature and Science in Tsukuba, Japan. The whole plant was separated into different parts as root, stem, leaves, and flower, and then dried in the shade for a week. A voucher specimen (2001-08) of the whole plant was registered and deposited at the herbarium of the National Museum of Nature and Science in Tsukuba, Japan as well as at the Department of Food Science and Nutrition, Pukyong National University (Professor J. S. Choi).

### 4.3. Extraction, Fractionation, and Isolation

The extraction, fractionation, and isolation of flavonoid compounds were carried out as previously described [16,20,50]. The extraction of dried leaves (1.28 kg) were carried out by refluxing with 2.5 L of MeOH for three times. Then, the MeOH extract (310.0 g) were partitioned to obtained solvent soluble fractions of CH_2_Cl_2_ (68.8 g), EtOAc (80.6 g), *n*-BuOH (30.7 g), and water (125.8 g). Furthermore, the repeated column chromatography of EtOAc fraction gave rise to the isolation of luteolin and its glycoside compounds. The obtained spectra were measured using JEOL JNM ECP-400 spectrometer (Tokyo, Japan) at 400 MHz for ^1^H NMR and 100 MHz for ^13^C NMR in deuterated dimethyl sulfoxide (DMSO-*d*_6_). Luteolin, luteolin 5-*O*-*β*-d-glucopyranoside, and luteolin 7-*O*-*β*-d-glucopyranoside were identified by spectroscopic methods, including ^1^H- and ^13^C-NMR, as well as by comparison with published spectral data [51,52] and thin layer chromatography analysis.

### 4.4. In Vitro α-Glucosidase Inhibitory Activity Assay

The inhibitory activity against yeast α-glucosidase enzyme was carried out spectrophotometrically in a 96-well micro-plate reader (Molecular Devices, Sunnyvale, CA, USA), using a procedure reported by Li et al. [53] with slight modifications. Shortly, a reaction mixture containing 100 mM phosphate buffer (pH 6.8), 20 µL of enzyme together with/without the various concentration of 20 µL of sample solutions/positive control (acarbose) were incubated at 37 °C for 5 min followed by the addition of 20 µL of *p*-NPG. After 15 min of incubation, the reaction was ceased using 80 µL of stop solution (0.2 M sodium carbonate) and absorbance was measured at 405 nm.

### 4.5. In Vitro Protein Tyrosine Phosphatase 1B (PTP1B) Inhibitory Activity Assay

The inhibitory activity of the isolated compounds against human recombinant PTP1B was evaluated using p-NPP as a substrate with slight modifications [54]. Shortly, a PTP1B reaction buffer containing 50 mM citrate buffer (pH 6.0), 0.1 M sodium chloride, 1 mM EDTA and 1 mM DTT, 10 µL of enzyme and 50 µL of *p*-NPP together with/without the various concentration of 10 µL of sample solutions/positive control (ursolic acid) were incubated at 37 °C for 15 min. After incubation, the reaction was ceased using 10 µL of stop solution (10 mM sodium hydroxide) and absorbance was measured at 405 nm.

### 4.6. In Vitro BACE1 Enzyme Assay

Each assay was carried out according to the supplied instructions provided by the manufacturer with slight modifications [55]. Shortly, a reaction mixture contatining 10 µL of 50 mM sodium acetate buffer (pH 4.5), 10 µL of BACE1 and 10 µL of substrate (750 nM Rh-EVNLDAEFK-Quencher in 50 mM amminium bicarbonate) together with/without the various concentration of 10 µL of sample solutions/positive control (quercetin) were incubated at 25 °C for 60 min in the dark. After incubation, the reaction was ceased using 10 µL of BACE1 stop solution and absorbance was measured at 545 nm (emmission) and 585 nm (emission) wavelengths using microplate spectrofluorometer (Spectramax Gemini XPS, Molecular Devices).

### 4.7. Pro-Oxidant Assay

The pro-oxidant activity of isolated compounds were assessed with slight modifications [56,57]. Shortly, 250 µL of sample solutions/positive control (ascorbic acid) dissolved in 0.2 M phosphate buffer (pH 6.6) and 25 µL of 1% potassium ferricyanide [K_3_Fe(CH_6_)] were incubated at 50 °C for 20 min. After that, an equal volume of 10% trichloroacetic acid was added and then centriguged at 3000 *g* for 10 min. The supernatant (500 µL) and distilled water (500 µL) was mixed followed by addition of 0.1% ferric chloride (0.1 mL). Immediately, absorbance were measured spectrophotometrically at 700 nm.

### 4.8. 2-Diphenyl-1-Picrylhydrazyl (DPPH) Radical Scavenging Activity

DPPH radical scavenging activity were carried out using procedures previously described with slight modifications [58]. Shortly, 160 µL of sample solutions/positive control (ascorbic acid) were mixed with 40 µL of DPPH solution and incubated at room temperature for 30 min. After that, absornace were measured at 520 nm using using a microplate reader spectrophotometer VERSAmax (Molecular Devices).

### 4.9. Peroxynitrite (ONOO^−^) Scavenging Activity

Peroxynitrite (ONOO^−^) scavenging activity was carried ou with slight modifications [59]. Shortly, 10 µL of sample solutions/positive control (penicillamine) were mixed with rhodamine buffer (pH 7.4) containing 50 mM sodium phosphate dibasic, 50 mM sodium phosphate monobasic, 90 mM sodium chloride, 5 mM potassium chloride, and 100 μM DTPA. The final DHR 123 concentration was 5 µM. The fluorescent intensities were measured after 5 min with/without authentic ONOO^−^ using a fluorescence microplate reader (Spectramax Gemini XPS, Molecular Devices) at wavelenghts of 485 and 530 nm of excitation and emission, respectively.

### 4.10. Kinetic Parameters of Active Compounds towards α-Glucosidase and BACE1 Inhibition

The inhibition constant (*K_i_*) and inhibition mode of α-glucosidase and BACE1 were calculated with Lineweaver-Burk and Dixon plots [60,61]. The kinetic parameters were obtained over various concentrations of substrate (0, 0.625, 1.25, and 2.5 mM for α-glucosidase and 375, 500, and 750 nM for BACE1) and inhibitors (0, 10, 30, and 60 µM for luteolin and 0, 100, 200, and 400 µM for luteolin 5-*O*-*β*-d-glucopyranoside and luteolin 7-*O*-*β*-d-glucopyranoside for α-glucosidase and 0, 10, 30, and 50 µM luteolin for BACE1). Each graph was generated using SigmaPlot 12.0 (Systat Software Inc., San Jose, CA, USA) and GraphPad Prism 5 (GraphPad Prism Software, Inc., La Jolla, CA, USA).

### 4.11. α-Glucosidase and BACE1 Molecular Docking Simulations

In order to find the behavior of the small molecule in the binding pockets of target proteins, molecular docking simulation was performed. For docking studies, the crystal structure of α-glucosidase and BACE1 protein targets were obtained from the RCSB Protein Data Bank with the respective accession codes 3A4A and 2WJO, respectively. The co-crystallized ligands, acarbose and (Z)-3-butylidenephthalide (BIP) for α-glucosidase and 2-amino-3-{(1*R*)-1-cyclohexyl-2-[(cyclohexyl-carbonyl)amino]ethyl}-6-phenoxyquinazolin-3-ium (QUD) and 3,5,7,3′,4′-pentamethoxyflavone (PMF) for BACE1, were used to generate the grid box for catalytic and allosteric inhibition mode respectively with compounds CIDs of 41774, 5352899, 44631815, and 97332, respectively. The 3D structures of the isolated compounds luteolin, luteolin 5-*O*-*β*-d-glucopyranoside, and luteolin 7-*O*-*β*-d-glucopyranoside were downloaded from PubChem Compound (NCBI), with compound CIDs of 5280445, 15559460, and 5280637, respectively. The results were visualized and analyzed using PyMOL 1.7.4 (Schrödinger, LLC, New York, NY, USA) and LigPlot+ 1.4.5 (European Bioinformatics Institute, London, UK).

### 4.12. Pharmacokinetic Profile of Active Compounds from C. maackii

The pharmacokinetics parameters, absorption, distribution, metabolism, and excretion (ADME) were evaluated using the open source tool PreADMET (https://preadmet.bmdrc.kr/adme/).

### 4.13. Statistical Analysis

Statistical significance was analyzed through Student’s *t*-test using Microsoft Excel 2016 (Microsoft Corporation, Redmond, WA, USA) and was noted at *p* < 0.05, *p* < 0.001, and *p* < 0.0001 in the Table 1, Table 2 and Table 3. All experiments were carried out in triplicate, repeated on three individual days and expressed as the mean ± SD (n = 3).

## 5. Conclusions

Among the various parts of the *C. maackii* plant, the leaves demonstrated notable activity towards α-glucosidase and BACE1, which minimizes the inconvenience of using the whole plant or other, less active parts for treatment. Moreover, the isolated compound luteolin showed concentration-dependent non-competitive inhibition against α-glucosidase and BACE1, which was in accordance with the molecular docking simulation studies. At the same time, its glucosides, luteolin 5-*O*-*β*-d-glucopyranoside and luteolin 7-*O*-*β*-d-glucopyranoside, both showed moderate activity. Thus, Korean thistle (*C. maackii*), which is widely used as a folk medicine, may act as an important dietary supplement in treating T2D and AD. The leaves of this plant are of particular interest because of the concentration of the active compound luteolin along with its glucosides, which are promising candidates for more detailed in vitro and in vivo studies.

## Figures and Tables

**Figure 1 molecules-24-00649-f001:**
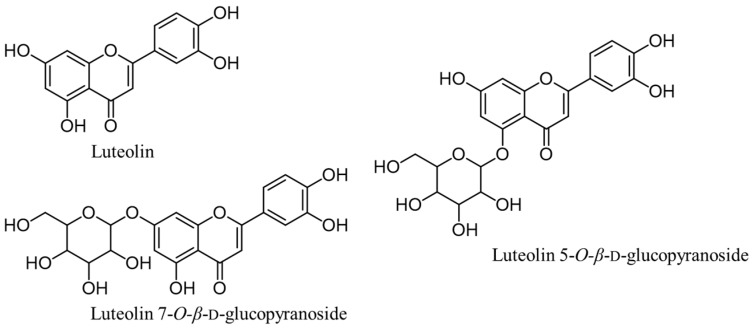
Structures of different isolated compounds from *C. maackii*.

**Figure 2 molecules-24-00649-f002:**
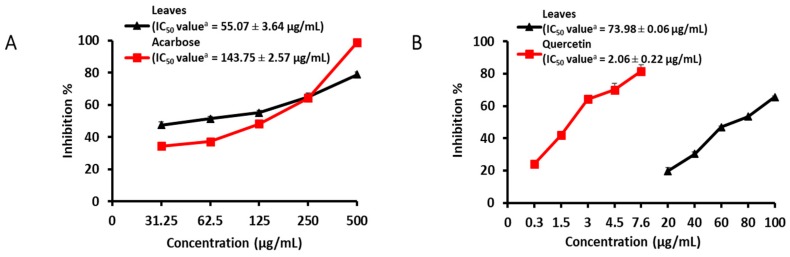
Concentration-dependent (**A**) α-glucosidase and (**B**) BACE1 inhibitory activity of MeOH extract of leaves of *C. maackii* along with the standards, acarbose and quercetin, respectively. Error bars indicate standard deviation (SD).

**Figure 3 molecules-24-00649-f003:**
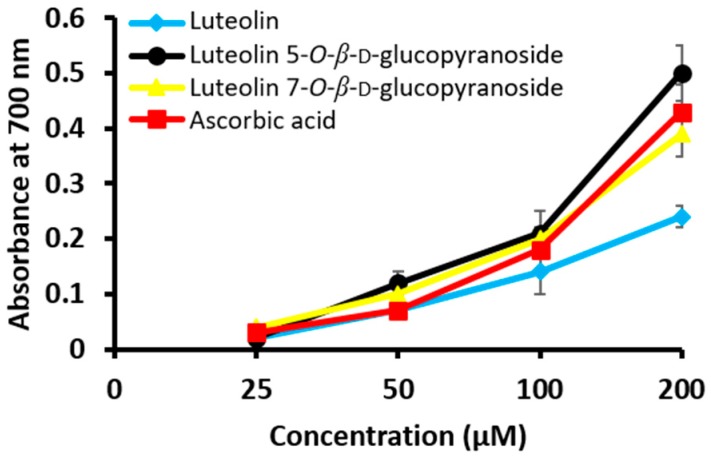
Pro-oxidant activity of different isolated compounds from *C. maackii* as measured from Fenton reaction using ascorbic acid as positive control. Data are represented as mean ± SD of triplicate experiments.

**Figure 4 molecules-24-00649-f004:**
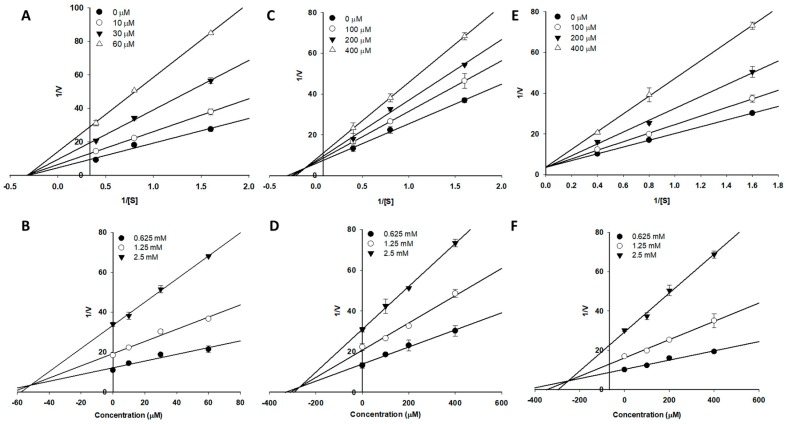
Lineweaver-Burk plot (**A**,**C**,**E**) and Dixon plot (**B**,**D**,**F**) for the inhibition of α-glucosidase by luteolin (**A**,**B**), luteolin 5-*O*-*β*-d-glucopyranoside (**C**,**D**), and luteolin 7-*O*-*β*-d-glucopyranoside (**E**,**F**), respectively.

**Figure 5 molecules-24-00649-f005:**
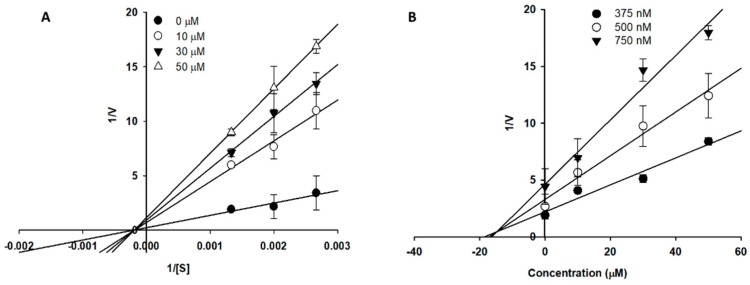
Lineweaver-Burk plot (**A**) and Dixon plot (**B**) for the inhibition of BACE1 by luteolin.

**Figure 6 molecules-24-00649-f006:**
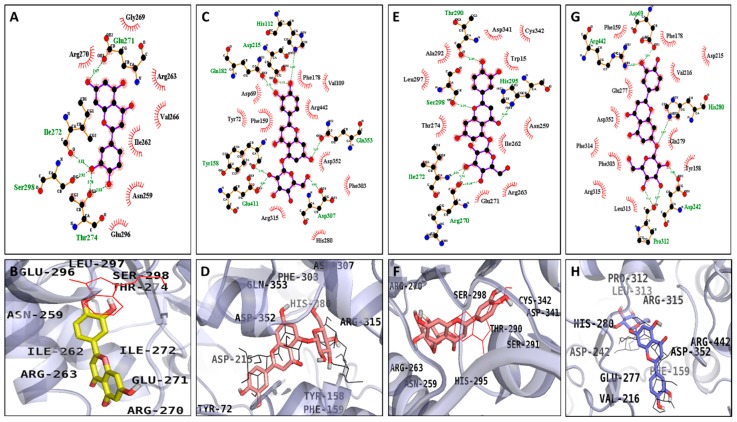
2D (**A**,**C**,**E**,**G**) and 3D (**B**,**D**,**F**,**H**) diagrams of α-glucosidase inhibition by luteolin (allosteric inhibition, **A**,**B**), luteolin 5-*O*-*β*-d-glucopyranoside (catalytic inhibition, **C**,**D**), luteolin 5-*O*-*β*-d-glucopyranoside (allosteric inhibition, **E**,**F**), and luteolin 7-*O*-*β*-d glucopyranoside (catalytic inhibition, **G**,**H**). Luteolin, luteolin 5-*O*-*β*-d glucopyranoside, luteolin 7-*O*-*β*-d-glucopyranoside, acarbose, and BIP are represented by yellow, bold red, blue, black, and red colored structures, respectively.

**Figure 7 molecules-24-00649-f007:**
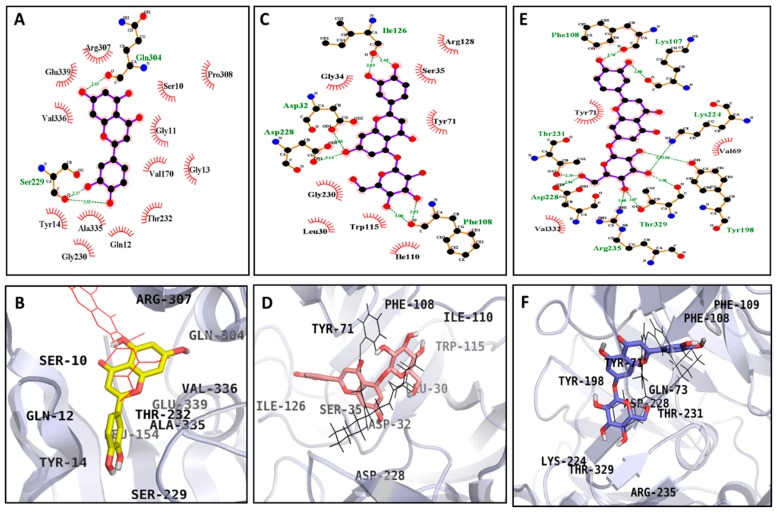
2D (**A**,**C**,**E**) and 3D (**B**,**D**,**F**) diagrams of BACE1 inhibition by luteolin (allosteric inhibition, **A**,**B**), luteolin 5-*O*-*β*-d-glucopyranoside (catalytic inhibition, **C**,**D**), and luteolin 7-*O*-*β*-d-glucopyranoside (catalytic inhibition, **E**,**F**). Luteolin, luteolin 5-*O*-*β*-d-glucopyranoside, luteolin 7-*O*-*β*-d-glucopyranoside, QUD, and PMF are represented by yellow, bold red, blue, black, and red colored structures, respectively.

**Table 1 molecules-24-00649-t001:** α-Glucosidase and BACE1 inhibitory activity of MeOH extract of different parts and solvent-soluble fractions of leaves of *C. maackii*.

Samples	α-Glucosidase		BACE1	
	Inhibition (%), Mean ± SD ^a^	IC_50_ Value ^b^	Inhibition (%), Mean ± SD ^a^	IC_50_ Value ^b^
Aerial part	70.21 ± 0.50 ***	375.66 ± 3.21	81.33 ± 1.44 ***	41.43 ± 0.23
MeOH extract stem	47.61 ± 1.29 **	>1000	16.84 ± 3.37 *	178.79 ± 2.41
MeOH extract root	16.66 ± 0.44 *	>1000	36.82 ± 4.47 **	>100
MeOH extract flower	9.93 ± 1.65 *	>1000	52.11 ± 0.66 ***	91.44 ± 0.84
MeOH extract leaves	78.98 ± 1.75 ***	55.07 ± 3.64	65.47 ± 0.65 ***	73.98 ± 0.06
CH_2_Cl_2_ fraction	11.59 ± 5.39	–	79.08 ± 0.36 ***	25.77 ± 0.57
EtOAc fraction	75.65 ± 3.61 ***	32.79 ± 1.63	93.01 ± 0.68 ***	39.88 ± 1.66
*n*-BuOH fraction	16.80 ± 2.01 *	–	35.7 ± 4.99 *	150.53 ± 3.95
H_2_O fraction	3.1 ± 1.7	–	*NA*	–
Acarbose ^c^/Quercetin ^c^	99.04 ± 0.20 ***	143.75 ± 2.57	70.01 ± 3.91 ***	2.06 ± 0.22
Control	0.00 ± 5.45		0.00 ± 5.60	

^a^ Mean ± standard deviation (SD) of three assays at a concentration of 500 and 100 µg/mL for MeOH extract + positive control and fractions, respectively, for α-glucosidase. And 100 µg/mL for BACE1 except CH_2_Cl_2_ fraction (50 µg/mL) and quercetin (4.5 µg/mL). ^b^ The 50% inhibition concentrations (IC_50_, µg/mL) are expressed as the mean ± SD of triplicates. ^c^ Positive controls for α-glucosidase and β-site amyloid precursor protein cleaving enzyme 1 (BACE1), respectively. * *p* < 0.05, ** *p* < 0.001, and *** *p* < 0.0001 indicate significance differences from control. *NA* No activity, (−) Not determined.

**Table 2 molecules-24-00649-t002:** Anti-oxidant activity of different isolated compounds from *C. maackii*.

Compounds	DPPH		ONOO^−^	
	Inhibition (%), Mean ± SD ^a^	EC_50_ Value ^b^	Inhibition (%), Mean ± SD ^a^	EC_50_ Value ^b^
Luteolin	76.65 ± 0.73 **	3.05 ± 0.06	85.21 ± 1.61 **	0.86 ± 0.04
Luteolin 5-*O*-*β*-d-gluco-pyranoside	52.43 ± 0.53 **	4.76 ± 0.05	79.23 ± 4.95 *	0.50 ± 0.03
Luteolin 7-*O*-*β*-d-gluco-pyranoside	58.97 ± 1.95 **	4.79 ± 0.19	80.23 ± 4.22 **	2.51 ± 0.05
Ascorbic acid ^c^	18.72 ± 0.44 **	21.35 ± 0.09	–	–
Penicillamine ^c^	–	–	56.57 ± 2.03 *	2.28 ± 0.02
Control	0.00 ± 1.18		0.00 ± 7.68	

^a^ Mean ± SD of three assays at a concentration of 5 µM. ^b^ The 50% effective concentrations (EC_50_, µM) are expressed as the mean ± SD of triplicates. ^c^ Positive controls for DPPH and ONOO^−^, respectively. * *p* < 0.001 and ** *p* < 0.0001 indicate significance differences from control, (−) Not determined.

**Table 3 molecules-24-00649-t003:** Anti-diabetic and anti-Alzheimer’s disease activity of luteolin and its derivatives.

Compounds	α-Glucosidase		PTP1B	BACE1	
	Inhibition (%), Mean ± SD ^a^	IC_50_ Value ^b^ (*K_i_* ^c^, Inhibition Mode ^d^)	IC_50_ Value ^b^	Inhibition (%), Mean ± SD ^a^	IC_50_ Value ^b^ (*K_i_* ^c^, Inhibition Mode ^d^)
Luteolin	69.12 ± 1.20 ***	51.27 ± 1.23 (52.04, NC)	>400	73.73 ± 0.57 ***	13.75 ± 0.26 (14.76, NC)
Luteolin 5-*O*-*β*-d-glucopyranoside	14.02 ± 1.45 *	270.03 ± 4.69 (271.80, M)	>400	17.53 ± 1.65 *	969.97 ± 7.83
Luteolin 7-*O*-*β*-d-glucopyranoside	25.53 ± 1.21 *	248.77 ± 2.56 (251.20, C)	>400	*NA*	–
Acarbose ^e^	40.36 ± 0.40 **	267.27 ± 2.29	–	–	–
Ursolic acid ^e^	–	–	16.73 ± 0.52	–	–
Quercetin ^e^	–	–	–	67.56 ± 4.10 ***	6.30 ± 0.05

C Competitive, NC Non-competitive, and M Mixed type inhibitions. ^a^ Mean ± SD of three assays at a concentration of 100 µM except quercetin (10 µM). ^b^ The 50% inhibition concentrations (IC_50_, µM) are expressed as the mean ± SD of triplicates. ^c^ Inhibition constants (*K_i_*) were determined using Dixon plot. ^d^ Inhibition types were determined using Lineweaver-Burk plot. ^e^ Positive controls. * *p* < 0.05, ** *p* < 0.001, and *** *p* < 0.0001 indicate significance differences from control values 0.00 ± 5.44 for α-glucosidase and 0.00 ± 3.84 for quercetin. *NA* No activity, (−) Not determined.

**Table 4 molecules-24-00649-t004:** Molecular interactions of the α-glucosidase (3A4A) active site with luteolin, luteolin 5-*O*-*β*-d-glucopyranoside and luteolin 7-*O*-*β*-d-glucopyranoside as well as reported inhibitors.

Compounds	Docking Score (kcal/mol)	No. of H-Bond	H-Bond Interacting Residues	Hydrophobic Interacting Residues
Luteolin (Allosteric)	−7.25	5	Glu271, Ile272, Ser298, Thr274	Gly269, Arg270, Glu296, Asn259, Ile262, Val266, Arg263
Luteolin 5-*O*-*β*-d-glucopyranoside (Catalytic)	−7.22	7	His112, Asp215, Gln182, Tyr158, Glu411, Asp307, Gln353	Asp69, Tyr72, Phe159, Arg315, His280, Asp352, Phe303, Arg442, Val109, Phe178
Luteolin 5-*O*-*β*-d-glucopyranoside (Allosteric)	−6.98	6	Thr290, Ser298, Ile272, Arg270, His295	Asp341, Cys342, Trp15, Asn259, Ile262, Arg263, Glu271, Thr274, Leu297, Ala292
Luteolin 7-*O*-*β*-d-glucopyranoside (Catalytic)	−7.21	6	Asp69, Arg442, Pro312, Asp242, His280	Phe159, Glu277, Asp352, Phe314, Phe303, Arg315, Leu313, Tyr158, Gln279, Val216, Asp215, Phe178
Acarbose ^a^ (Catalytic inhibitor)	−8.59	17	Gln182, Asp69, Asp215, Arg213, Glu277, Asp352, Arg442, Asp307, His280, Asp242, Ser240, Tyr158	Lys156, Arg315, Gln279, Phe178, Phe303, Gln353, Tyr72, Val216, His351, Glu411
BIP ^b^ (Allosteric inhibitor)	−6.89	2	Glu296, His295	Asp341, Cys342, Ala292, Arg294, Leu297, Ser291, Asn259, Thr290, Ser298, Trp15, Lys16, Trp343

^a^ Reported competitive inhibitor. ^b^ Reported allosteric inhibitor.

**Table 5 molecules-24-00649-t005:** Molecular interactions of the BACE1 (2WJO) active site with luteolin, luteolin 5-*O*-*β*-d-glucopyranoside and luteolin 7-*O*-*β*-d-glucopyranoside as well as reported inhibitors.

Compounds	Docking Score (kcal/mol)	No. of H-Bond	H-Bond Interacting Residues	Hydrophobic Interacting Residues
Luteolin (Allosteric)	−7.18	3	Ser299, Gln304	Ser10, Gly11, Gln12, Gly13, Pro308, Val170, Thr232, Gly230, Ala335, Tyr14, Val336, Glu339, Arg307
Luteolin 5-*O*-*β*-d-glucopyranoside (Catalytic)	−6.02	6	Asp32, Asp228, Ile126, Phe108	Leu30, Gly34, Ser35, Tyr71, Ile110, Trp115, Gly230, Arg128
Luteolin 7-*O*-*β*-d-glucopyranoside (Catalytic)	−5.47	9	Lys107, Phe108, Lys224, Tyr198, Thr329, Arg235, Asp228, Thr231	Tyr71, Val69, Val332
QUD ^a^ (Catalytic inhibitor)	−9.3	4	Asp32, Asp228, Gly230	Lys75, Gly74, Leu30, Thr231, Val69, Tyr198, Ile226, Thr329, Gly34, Ser35, Arg235, Tyr71, Ile118, Lys107
PMF ^b^ (Allosteric inhibitor)	−6.15	1	Ser10	Ala168, Glu339, Val170, Thr232, Gly11, Gln304, Gly156, Ala335, Arg307, Pro308, Ala157

^a^ Reported competitive inhibitor. ^b^ Reported allosteric inhibitor.

**Table 6 molecules-24-00649-t006:** Absorption, distribution, metabolism, and excretion (ADME) properties of luteolin and its derivatives from *C. maackii*.

Compounds	Molecular Weight (g/mol)	% HIA ^a^	Plasma Protein Binding	MDCK ^b^	Caco-2 Cell
Luteolin	286.24	79.43	99.71	36.52	4.54
Luteolin 5-*O*-*β*-d-glucopyranoside	448.38	25.17	68.05	0.70	3.12
Luteolin 7-*O*-*β*-d-glucopyranoside	448.38	25.16	73.28	0.75	4.87

^a^ Percent human intestinal absorption. ^b^ Madin-Darby canine kidney.

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
