# Peer review of "Korean Thistle (Cirsium japonicum var. maackii (Maxim.) Matsum.): A Potential Dietary Supplement against Diabetes and Alzheimer’s Disease"

_molecules, 2019, doi:10.3390/molecules24030649_

Round 1

Reviewer 1 Report

The authors described the effects of  luteolin, obtained from extracts of  a plant named Cirsium japonicum var. maackii on both alpha-glicosidase and β-54 site amyloid precursor protein cleaving enzyme 1 (BACE1) key enzymes associated with Type2 Diabetes and Alzheimer disease (AD) respectively. Indeed these results suggesting that luteolin should be effective compound against diabetes and AD.

 However, some criticisms persist.

Both T2D and AD are usually progressive pathologies with functional deterioration, i.e. it is still unclear how the level of transcriptional

changes that usually occur in the liver in insulin resistance should be associated to luteolin response.

 So based on the antioxidant and prooxidant properties of luteolin the potential effect of luteolin can be considered speculative on the light of the reported results.

Author Response

Dear Reviewer,

We would like to thank for your valuable comments regarding the manuscript entitled “Cirsium japonicum var. maackii (Maxim.) Matsum., Korean Thistle: A Potential Dietary Supplement against Diabetes and Alzheimer’s Disease”. Your suggestions led us to an improvement of the work. We have corrected the manuscript according to suggestion and a copy of the revised manuscript has been attached where the revisions are highlighted by the red color for the convenience. We hope our correction has improved the manuscript to a level of their satisfaction.

Reviewer #1:

The authors described the effects of luteolin, obtained from extracts of a plant named Cirsium japonicum var. maackii on both alpha-glicosidase and β-54 site amyloid precursor protein cleaving enzyme 1 (BACE1) key enzymes associated with Type2 Diabetes and Alzheimer disease (AD) respectively. Indeed, these results suggesting that luteolin should be effective compound against diabetes and AD.

 However, some criticisms persist.

Both T2D and AD are usually progressive pathologies with functional deterioration, i.e. it is still unclear how the level of transcriptional

changes that usually occur in the liver in insulin resistance should be associated to luteolin response.

 So based on the antioxidant and prooxidant properties of luteolin the potential effect of luteolin can be considered speculative on the light of the reported results.

Answer: According to the suggestions provided, the pro-oxidant and anti-oxidant activity of isolated compounds have been added in the revised manuscript as Figure 3 and Table 2 in result section 2.2 and 2.3 as:

“2.2. Pro-oxidant Activity of Different Compounds from C. maackii

The isolated compounds were tested against pro-oxidant activity which were depicted in Figure 3. Our result demonstrated that luteolin have lower reducing power than positive control ascorbic acid. While in case of its glucosides, luteolin 5-O-β-d-glucopyranoside exhibited higher reducing power.

2.3. Anti-Oxidant Activity of Different Compounds from C. maackii

Among the isolated compounds, the anti-oxidant activity [2,2-diphenyl-1-picrylhydrazyl (DPPH) radical scavenging and perxoynitrite (ONOO) scavenging activity] were carried out as shown in Table 2. All three compounds showed significant inhibition of DPPH radical when compared to that of positive control, ascorbic acid (21.35 ± 0.09 µM). Additionally, luteolin (3.05 ± 0.06 µM), an aglycoside moiety exhibited powerful DPPH scavenging ability than its glycosides. Moreover, for ONOO scavenging activity, both luteolin and luteolin 5-O-β-d-glucopyranoside exhibited significant scavenging activity with half maximal effective concentration (EC50) values of 0.86 ± 0.04 and 0.50 ± 0.03 µM, respectively. Penicillamine, the positive control used for the experiment showed an EC50 of 2.28 ± 0.02 µM”.

Reviewer 2 Report

Dear Authors,

I do find this study interesting and publishable. I actually thought to rate it as high in all ratings categories above, but I decided finally for average as this is in vitro study and there is no evidence yet that the in vivo effect will be similar. The work is a step forward in the field, and animal model experimentation should be the next step and should include toxicity studies and dose-response relationship.

Best wishes,

Author Response

Dear Reviewer,

We would like to thank for your valuable comments regarding the manuscript entitled “Cirsium japonicum var. maackii (Maxim.) Matsum., Korean Thistle: A Potential Dietary Supplement against Diabetes and Alzheimer’s Disease”. Your suggestions led us to an improvement of the work. We have corrected the manuscript according to suggestion and a copy of the revised manuscript has been attached where the revisions are highlighted by the red color for the convenience. We hope our correction has improved the manuscript to a level of their satisfaction.

Reviewer 2

Dear Authors,

I do find this study interesting and publishable. I actually thought to rate it as high in all ratings categories above, but I decided finally for average as this is in vitro study and there is no evidence yet that the in vivo effect will be similar. The work is a step forward in the field, and animal model experimentation should be the next step and should include toxicity studies and dose-response relationship.

Best wishes,

Answers: We would like to thank you for your wonderful comments and suggestions regarding the manuscript. Considering your advice, our next focus will definitely be in the animal experiment in order to emphasized the importance of the plant Cirsium maackii and we have mentioned in the manuscript as well as, “Nevertheless, additional in vivo studies are needed to strengthened the in vitro results” (Line 368-369).

Reviewer 3 Report

Title:Cirsium japonicum var. Maackii (Maxim.) Matsum., Korean Thistle: A Potential Dietary Supplement 3 against Diabetes and Alzheimer’s Disease

Authors: Aditi Wagle, Su Hui Seong, Srijan Shrestha, Hyun Ah Jung, and Jae Sue Choi

Manuscript#:molecules-434489

Summary:In this study Wagle et al. has shown the potential of having inhibitory activity against α-glucosidase and BACE1 of Cirsium japonicum var. maacki(Maxim.) Matsum, thus can be used in Diabetes and Alzheimer’s Disease. They have used molecular docking and enzyme kinetic analysis approach to prove that the extract has the potential to be used as dietary supplement against Diabetes and Alzheimer’s Disease.

Critique: Even though the authors have shown the potential of the extract of Cirsium japonicum var. maacki(Maxim.) Matsum, the manuscript cannot be accepted in its current form. Following changes are warranted.

1)   The authors should incorporate the in-vitro α-glucosidase, BACE1 and PTP1B inhibitory assay briefly. They have cited the papers from which they have adapted the assay, but it is advisable to have very brief description of the assay in the manuscript.

2)    The same should be done for extraction procedures.

3)   The authors should mention the reason the need to make the extracts in the different solvents as mentioned in the table 1. And they should mention the solvent they used to make luteolin and the luteolin derivatives. The reason it is important that all the extract made in different solvents had different IC50 value and activity towards the enzymes.

Author Response

Dear Reviewer,

We would like to thank for your valuable comments regarding the manuscript entitled “Cirsium japonicum var. maackii (Maxim.) Matsum., Korean Thistle: A Potential Dietary Supplement against Diabetes and Alzheimer’s Disease”. Your suggestions led us to an improvement of the work. We have corrected the manuscript according to suggestion and a copy of the revised manuscript has been attached where the revisions are highlighted by the red color for the convenience. We hope our correction has improved the manuscript to a level of their satisfaction.

Reviewer #3:

Title: Cirsium japonicum var. maackii (Maxim.) Matsum., Korean Thistle: A Potential Dietary Supplement 3 against Diabetes and Alzheimer’s Disease

Authors: Aditi Wagle, Su Hui Seong, Srijan Shrestha, Hyun Ah Jung, and Jae Sue Choi

Manuscript#:molecules-434489

Summary: In this study Wagle et al. has shown the potential of having inhibitory activity against α-glucosidase and BACE1 of Cirsium japonicum var. maackii (Maxim.) Matsum, thus can be used in Diabetes and Alzheimer’s Disease. They have used molecular docking and enzyme kinetic analysis approach to prove that the extract has the potential to be used as dietary supplement against Diabetes and Alzheimer’s Disease.

Critique: Even though the authors have shown the potential of the extract of Cirsium japonicum var. maacki (Maxim.) Matsum, the manuscript cannot be accepted in its current form. Following changes are warranted.

 1)   The authors should incorporate the in-vitro α-glucosidase, BACE1 and PTP1B inhibitory assay briefly. They have cited the papers from which they have adapted the assay, but it is advisable to have very brief description of the assay in the manuscript.

Answer: The brief description of the assay procedures has been added in the sections 4.4, 4.5, and 4.6 of the revised manuscript (Line 437-460) as:

“4.4. In Vitro α-Glucosidase Inhibitory Activity Assay

The inhibitory activity against yeast α-glucosidase enzyme was carried out spectrophotometrically in a 96-well micro-plate reader (Molecular Devices, LLC), using a procedure reported by Li et al. [53] with slight modifications. Shortly, a reaction mixture containing 100 mM phosphate buffer (pH 6.8), 20 µL of enzyme together with/without the various concentration of 20 µL of sample solutions/positive control (acarbose) were incubated at 37 ℃ for 5 minutes followed by the addition of 20 µL of p-NPG. After 15 minutes of incubation, the reaction was ceased using 80 µL of stop solution (0.2 M sodium carbonate) and absorbance was measured at 405 nm.

4.5. In Vitro Protein Tyrosine Phosphatase 1B (PTP1B) Inhibitory Activity Assay

The inhibitory activity of the isolated compounds against human recombinant PTP1B was evaluated using p-NPP as a substrate with slight modifications [54]. Shortly, a PTP1B reaction buffer containing 50 mM citrate buffer (pH 6.0), 0.1 M sodium chloride, 1 mM EDTA and 1 mM DTT, 10 µL of enzyme and 50 µL of p-NPP together with/without the various concentration of 10 µL of  sample solutions/positive control (ursolic acid) were incubated at 37 ℃ for 15 minutes. After incubation, the reaction was ceased using 10 µL of stop solution (10 mM sodium hydroxide) and absorbance was measured at 405 nm.

4.6. In Vitro BACE1 Enzyme Assay

Each assay was carried out according to the supplied instructions provided by the manufacturer with slight modifications [55]. Shortly, a reaction mixture contatining 10 µL of 50 mM sodium acetate buffer (pH 4.5), 10 µL of BACE1 and 10 µL of substrate (750 nM Rh-EVNLDAEFK-Quencher in 50 mM amminium bicarbonate) together with/without the various concentration of 10 µL of  sample solutions/positive control (quercetin) were incubated at 25 ℃ for 60 minutes in the dark. After incubation, the reaction was ceased using 10 µL of BACE1 stop solution and absorbance was measured at 545 nm (emmission) and 585 nm (emission) wavelengths using microplate spectrofluorometer (Spectramax Gemini XPS, Molecules devices)”.

2)    The same should be done for extraction procedures.

Answer: The short description of the extraction procedure has been added in the 4.3. section (Line 426-433) as:

4.3. Extraction, Fractionation, and Isolation

The extraction, fractionation, and isolation of flavonoid compounds were carried out as previously described [16,20,50]. The extraction of dried leaves (1.28 kg) were carried out by refluxing with 2.5 L of MeOH for three times. Then, the MeOH extract (310.0 g) were partitioned to obtained solvent soluble fractions of CH2Cl2 (68.8 g), EtOAc (80.6 g), n-BuOH (30.7 g), and water (125.8 g). Furthermore, the repeated column chromatography of EtOAc fraction gave rise to the isolation of luteolin and its glycoside compounds. The obtained spectra were measured using JEOL JNM ECP-400 spectrometer (Tokyo, Japan) at 400 MHz for 1H NMR and 100 MHz for 13C NMR in deuterated dimethyl sulfoxide (DMSO-d6). Luteolin, luteolin 5-O-β-d-glucopyranoside, and luteolin 7-O-β-d-glucopyranoside were identified by spectroscopic methods, including 1H- and 13C-NMR, as well as by comparison with published spectral data [51,52] and thin layer chromatography analysis”.

3)   The authors should mention the reason the need to make the extracts in the different solvents as mentioned in the table 1. And they should mention the solvent they used to make luteolin and the luteolin derivatives. The reason it is important that all the extract made in different solvents had different IC50 value and activity towards the enzymes.

Answer: As in an extract there are mixtures of different polar and non-polar components. And partitioning of the extract will help to separate the polar and non-polar components into their solvent soluble fractions. This will make easy to find of the particular active portion responsible for the different activity which was exerted by the samples. In the manuscript, page 11, line 265-271, we explained the reason behind the partition of the MeOH extract as,

 “Furthermore, the high activity of the leaves due to their abundance of active constituents encourages us to find out which fractions to be responsible for their activity. Therefore, the partitioning of the MeOH extract leads to the four solvent soluble fractions where the separation of the components was based on their polarity. And among the four fractions, the EtOAc fraction showed the significant inhibitory activity towards α-glucosidase; while the CH2Cl2, followed by the EtOAc and the n-BuOH fractions, showed the most potent inhibition of BACE1”.

Round 2

Reviewer 1 Report

The revised manuscript has certainly improved based on the reviewer comments. The responses to referee sound with their suggestions.